# Treatment Settings and Outcomes with Regorafenib and Trifluridine/Tipiracil at Third-Line Treatment and beyond in Metastatic Colorectal Cancer: A Real-World Multicenter Retrospective Study



Carlo Signorelli [1,*], Maria Alessandra Calegari [2], Michele Basso [2], Annunziato Anghelone [2], Jessica Lucchetti [3], Alessandro Minelli [3], Lorenzo Angotti [3], Ina Valeria Zurlo [4], Marta Schirripa [1], Mario Giovanni Chilelli [1], Cristina Morelli [5], Emanuela Dell'Aquila [6], Antonella Cosimati [6], Donatello Gemma [7], Marta Ribelli [8], Alessandra Emiliani [8], Domenico Cristiano Corsi [8], Giulia Arrivi [9], Federica Mazzuca [9], Federica Zoratto [10], Maria Grazia Morandi [11], Fiorenza Santamaria [12,13], Rosa Saltarelli [14] and Enzo Maria Ruggeri [1]

1 Medical Oncology Unit, Belcolle Hospital, ASL Viterbo, 01100 Viterbo, Italy
2 Unit of Medical Oncology, Comprehensive Cancer Center, Fondazione Policlinico Universitario Agostino Gemelli, IRCCS, 00168 Rome, Italy
3 Division of Medical Oncology, Policlinico Universitario Campus Bio-Medico, 00128 Rome, Italy
4 Medical Oncology, "Vito Fazzi" Hospital, 73100 Lecce, Italy
5 Medical Oncology Unit, Department of Systems Medicine, Tor Vergata University Hospital, 00133 Rome, Italy
6 Medical Oncology 1, IRCCS Regina Elena National Cancer Institute, 00144 Rome, Italy
7 Medical Oncology Unit, ASL Frosinone, 03039 Sora (FR), Italy
8 Medical Oncology Unit, Ospedale San Giovanni Calibita Fatebenefratelli, Isola Tiberina, Gemelli Isola, 00186 Rome, Italy
9 Department of Clinical and Molecular Medicine, Sapienza University of Rome, Oncology Unit, Sant' Andrea Hospital, 00189 Rome, Italy
10 Medical Oncology Unit, ASL Latina, 04100 Latina, Italy
11 Medical Oncology Unit, San Camillo de Lellis Hospital, ASL Rieti, 02100 Rieti, Italy
12 Department of Experimental Medicine, Sapienza University of Rome, 00185 Rome, Italy
13 Medical Oncology A, Department of Radiological, Oncological and Pathological Sciences, Policlinico Umberto I, Sapienza University of Rome, 00185 Rome, Italy
14 UOC Oncology, San Giovanni Evangelista Hospital, ASL RM5, 00019 Tivoli (RM), Italy
* Correspondence: carlo.signorelli@asl.vt.it

**Abstract:** Background: Patients with refractory mCRC rarely undergo third-line or subsequent treatment. This strategy could negatively impact their survival. In this setting, regorafenib (R) and trifluridine/tipiracil (T) are two key new treatment options with statistically significant improvements in overall survival (OS), progression-free survival (PFS), and disease control with different tolerance profiles. This study aimed to retrospectively evaluate the efficacy and safety profiles of these agents in real-world practice. Materials and Methods: In 2012–2022, 866 patients diagnosed with mCRC who received sequential R and T (T/R, n = 146; R/T, n = 116]) or T (n = 325) or R (n = 279) only were retrospectively recruited from 13 Italian cancer institutes. Results: The median OS is significantly longer in the R/T group (15.9 months) than in the T/R group (13.9 months) ($p = 0.0194$). The R/T sequence had a statistically significant advantage in the mPFS, which was 8.8 months with T/R vs. 11.2 months with R/T ($p = 0.0005$). We did not find significant differences in outcomes between groups receiving T or R only. A total of 582 grade 3/4 toxicities were recorded. The frequency of grade 3/4 hand-foot skin reactions was higher in the R/T sequence compared to the reverse sequence (37.3% vs. 7.4%) ($p = 0.01$), while grade 3/4 neutropenia was slightly lower in the R/T group than in the T/R group (66.2% vs. 78.2%) ($p = 0.13$). Toxicities in the non-sequential groups were similar and in line with previous studies. Conclusions: The R/T sequence resulted in a significantly longer OS and PFS and improved disease control compared with the reverse sequence. R and T given not sequentially have similar impacts on survival. More data are needed to define the best sequence and to explore the efficacy of sequential (T/R or R/T) treatment combined with molecular-targeted drugs.

**Keywords:** metastatic colorectal cancer; regorafenib; trifluridine/tipiracil; third-line therapy; real-world study

## 1. Introduction

The third most prevalent cancer diagnosis worldwide is colorectal cancer (CRC). Every year, more than 50,000 people in the United States die with a diagnosis of CRC, and approximately 150,000 more receive a new diagnosis of CRC [1,2]. Although screening and lifestyle changes are credited with a drop in the overall incidence of CRC among older persons in recent decades, the reported incidence among younger adults has risen. The reported 5-year overall survival (OS) of metastatic CRC (mCRC) is approximately 15 percent [3], and metastases will appear in 33% of cases of CRC, either at presentation or during follow-up [4,5]. The growing number of possible treatments for mCRC, together with the use of some drugs in more than one line or as adjuvant therapy, can give the impression that the therapeutic landscape is complicated. The choice of treatment is influenced by a number of variables, including the molecular characteristics of the tumor, the treatment target, the general health of the patient, the tumor load, and the clinical course. The treatment of refractory CRC is a dynamic field [6–8]. Molecularly and non-molecularly selected patients will benefit from additional therapies. Standard fluorouracil (FU)-based chemotherapy was the mainstay of treatment before the development of targeted therapies for particular molecular subtypes and primary tumor sidedness. Third-line or further treatments are infrequently used in cases of refractory mCRC, and OS might suffer as a result. Regorafenib, trifluridine/tipiracil, and anti-epidermal growth factor receptor (EGFR) agents are all suggested as third-line treatments for RAS- and BRAF wild-type (wt) tumors in patients who have not previously received EGFR antibodies [9–12]. Regorafenib (R) is a multikinase inhibitor that is active against a number of angiogenic receptor tyrosine kinases (RTKs), oncogenic RTKs, stromal RTKs, and intracellular signaling kinases. Trifluridine/tipiracil (T) is a combination of trifluridine, a thymidine-based nucleic acid analog, and tipiracil hydrochloride, a thymidine phosphorylase inhibitor that promotes the maintenance of a high trifluridine concentration [13–17]. A recent study reported significantly longer OS and progression-free survival (PFS) with improved disease control after treatment with T plus bevacizumab compared with T alone, which may represent a new standard of care for the treatment of refractory mCRC that has progressed after two lines of therapy [18]. To date, in randomized phase III trials, T and R have been the main novel therapies associated with statistically significant improvements in OS, PFS, and disease control [19–24]. However, because of differences among cases and the presence of several molecular subtypes, evaluating therapy alternatives is difficult, and in the setting of varying tolerance profiles against minor improvements in overall survival and progression-free survival, T and R are still regarded by some as clinically unimportant [25–28]. The best protocols for the administration of T and R in patients with advanced disease, the identification of subpopulations that might benefit from vital treatment with these drugs, and the identification of those subpopulations that might benefit from treatment with these drugs vs. the best supportive therapy alone are all questions that have yet to be resolved [29–35]. Furthermore, T and R have not been directly compared to one another, and it is uncertain which agent should be given first [36–41]. This multi-institutional retrospective study aimed to assess the effect of R and T, administered sequentially or not, on clinical outcomes and safety in patients with mCRC resistant to conventional chemotherapies.

## 2. Patients and Methods

This 10-year (2012 to 2022) retrospective observational study was carried out in 13 Italian cancer centers. The inclusion criteria were as follows:

1.  Patients who have progressed following exposure to at least two prior regimens of standard chemotherapy using fluoropyrimidine, irinotecan, oxaliplatin, anti-vascular

endothelial growth factor (VEGF) antibodies (bevacizumab and aflibercept), or anti-EGFR antibodies (cetuximab or panitumumab);

2. Age > 18 years;
3. Eastern Cooperative Oncology Group performance status (ECOG PS) 0–2;
4. Known RAS mutation status;
5. Adequate organ function at the start of treatment;
6. Histologically confirmed stage IV adenocarcinoma of the colon or rectum with unresectable metastatic disease.

Gender, age at the time of the third line of treatment, surgery, molecular subtypes (RAS/BRAF mutations, mismatch repair, and/or microsatellite instability status), primary tumor, site of metastasis, prior chemotherapy, and targeted therapies were all considered as baseline variables. Additional clinical data, including ECOG PS, dates of introduction and discontinuation of R or T, the reason for discontinuation, best tumor response, PFS, OS, dose reduction, and treatment-related grade 3/4 adverse events (AEs; National Cancer Institute Common Terminology Criteria for Adverse Events (CTCAE), version 4.0), were retrospectively collected. The study was approved by the ethics committee (2022-no.1021/CE Lazio 1) and was carried out in conformity with the Declaration of Helsinki. To protect sensitive information, all data were anonymized, and patients were identified only by their initials and a number. In accordance with the law, the lead investigator served as the data manager and had access to the complete database.

The study design is shown in Figure 1. The patients were divided into four groups based on whether they received sequential treatment, R/T or T/R, T or R alone. The groups were compared based on the type of treatment.

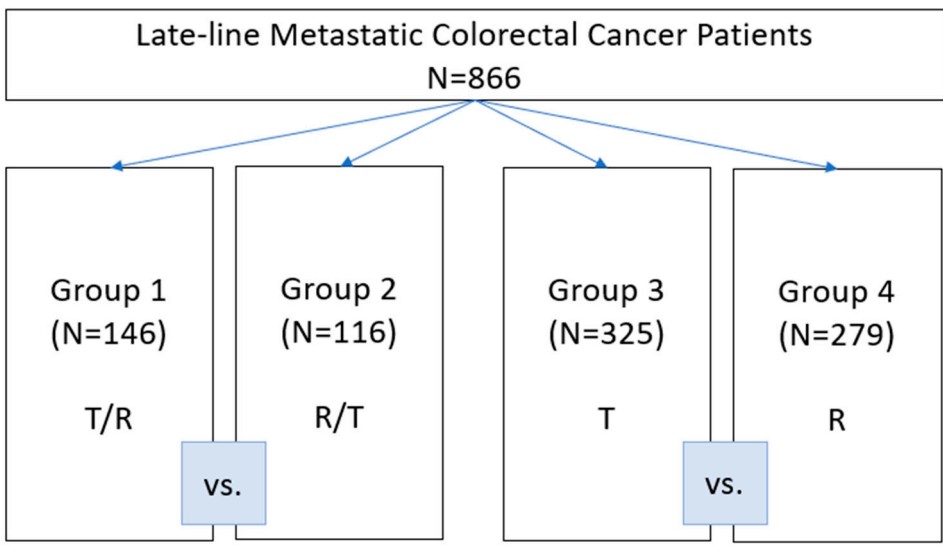

**Figure 1.** Study design.

The primary endpoints were overall survival (OS), defined as the time interval from the start of treatment until death from any cause (in the sequential treatment groups, OS was defined as the time between the first treatment, R or T, and death during the second treatment, T or R), and progression-free survival (PFS), defined as the time interval from the start of treatment until disease progression or death (in the sequential treatment groups, PFS was defined as the time between the first treatment, T or R, and disease progression or death during the second treatment, R or T). Living patients were censored at the last date when they were known to be alive.

The secondary objectives were the disease control rate (DCR), calculated as the sum of the rates of complete, partial, and stable disease, assessed using RECIST criteria, and compared according to whether the two drugs were administered sequentially or not,

and the objective response rate (ORR), which was the percentage of patients achieving an objective response (complete response or partial responses obtained in the sequential setting or not) according to RECIST criteria (version 1.1).

All patients receiving R or T or sequential R/T and T/R were included in the current study to remove potential selection bias. A predetermined investigator who was blind to the study's results and outcomes was responsible for selecting the patients. Patients were not chosen by the principal investigator who was in charge of statistical analyses.

To reduce the possibility of distortion bias, endpoints were predetermined at the time of the studies. We should be aware that the provided results should be viewed as exploratory because the study was retrospective.

Due to the nature of the retrospective investigation, informed permission was waived, and patient data confidentiality was maintained.

Trifluridine/tipiracil was given orally twice daily at a dose of 35 mg/m$^2$ on days 1–5 and 8–12 with 2 days off, for 2 weeks, followed by a 14-day rest period at the end of each month. Regorafenib was given as a standard dose of 160 mg once daily for 21 days of a 28-day cycle. The ReDos dose-escalation strategy of R (starting dose 80 mg/day orally with weekly escalation, per 40 mg increment, to 160 mg/day, permitted if there are no significant drug-related adverse events) was used according to the physician's decision [9,32]. Treatment was continued until disease progression, intolerable side effects, or the investigator's decision if considered clinically necessary; dose reductions were implemented based on toxicity and physician decision.

*Statistical Analysis*

The pertinent data were compiled using descriptive statistics. Possible relationships were assessed using the chi-square and Fisher-exact tests. The Kaplan–Meier product-limit approach was used to compute both PFS and OS and the log-rank test was used to evaluate differences between subgroups. Significance was established at $p = 0.05$. All of the statistical analyses were performed using SPSS Statistics software version 21.0.

## 3. Results

### 3.1. Patient Characteristics

The baseline patients' characteristics were well-balanced and are summarized in Table 1. A total of 866 consecutive patients with mCRC receiving sequential T and R (T/R (T prior to R), n = 146; R/T (R prior to T), n = 116)) or T (n = 325) or R (n = 279) alone were included in the analysis.

**Table 1.** Patients' characteristics at baseline.

|  | T/R | R/T | *p*-Value | T | R | *p*-Value |
|---|---|---|---|---|---|---|
|  | N (%) | N (%) |  | N (%) | N (%) |  |
| Total | 146 (16.8) | 116 (13.3) |  | 325 (37.5) | 279 (32.2) |  |
| Median age (years) | 69 (-) | 66 (-) |  | 71 (-) | 65 (-) |  |
| Patients ≥ 70yrs |  |  |  |  |  |  |
|     yes | 67 (45.8) | 42 (36.2) | 0.1306 | 178 (54.7) | 97 (34.7) | <0.00001 |
|     no | 79 (54.1) | 74 (63.7) |  | 147 (45.2) | 182 (65.2) |  |
| Gender |  |  |  |  |  |  |
|     Female | 51 (34.9) | 44 (37.9) | 0.6981 | 149 (45.8) | 118 (42.2) | 0.4115 |
|     Male | 95 (65.0) | 72 (62.0) |  | 176 (54.1) | 161 (57.7) |  |
| RAS status |  |  |  |  |  |  |
|     Wild type | 49 (33.5) | 42 (36.2) | 0.5986 | 122 (37.5) | 122 (43.7) | 0.1807 |
|     Mutant type | 90 (61.6) | 67 (57.7) |  | 195 (60.0) | 154 (55.1) |  |

**Table 1.** *Cont.*

|  | T/R | R/T | *p*-Value | T | R | *p*-Value |
|---|---|---|---|---|---|---|
| Tumor location |  |  |  |  |  |  |
| Right side | 52 (35.6) | 34 (29.3) | 0.2926 | 100 (30.7) | 91 (32.6) | 0.6611 |
| Left side | 94 (64.3) | 82 (70.6) |  | 225 (69.2) | 188 (67.3) |  |
| MSI |  |  |  |  |  |  |
| yes | 3 (2.0) | 4 (3.4) | 0.4594 | 3 (0.9) | 3 (1.0) | 0.675 |
| no | 89 (60.9) | 64 (55.1) |  | 177 (54.4) | 105 (37.6) |  |
| ECOG PS |  |  |  |  |  |  |
| 0–1 | 128 (87.6) | 108 (93.1) | 0.2113 | 250 (76.9) | 224 (80.2) | 0.3228 |
| 2 | 18 (12.3) | 8 (6.8) |  | 75 (23.0) | 55 (19.7) |  |
| Prior adjuvant therapy |  |  |  |  |  |  |
| yes | 46 (31.5) | 43 (37.0) | 0.232 | 90 (27.6) | 52 (18.6) | 0.0094 |
| no | 100 (68.4) | 66 (56.8) |  | 235 (72.3) | 227 (81.3) |  |
| Metastatic disease sites |  |  |  |  |  |  |
| Liver only | 23 (15.7) | 14 (12.0) | 0.4418 | 38 (11.6) | 31 (11.1) | 0.4508 |
| Liver + other | 80 (54.7) | 51 (43.9) |  | 133 (40.9) | 156 (55.9) |  |
| Others | 43 (29.4) | 51 (43.9) |  | 154 (47.3) | 92 (32.9) |  |
| CT 1°line regimen |  |  |  |  |  |  |
| Doublet chemotherapy | 112 (76.7) | 94 (81.0) | 0.4008 | 265 (81.5) | 140 (50.1) | 0.6262 |
| Triplet chemotherapy | 16 (10.9) | 9 (7.7) |  | 33 (10.1) | 14 (5.0) |  |
| CT 2°line regimen |  |  |  |  |  |  |
| Doublet chemotherapy | 119 (81.5) | 94 (81.0) | 1 | 258 (79.3) | 119 (42.6) | 0.3015 |
| Triplet chemotherapy | 2 (1.3) | 2 (1.7) |  | 5 (1.5) | 5 (1.7) |  |
| CT 3°line regimen |  |  |  |  |  |  |
| Fluoropyrimidine alone | 3 (2.0) | 2 (1.7) | 0.6221 | 4 (1.2) | 8 (2.8) | 0.3334 |
| Doublet chemotherapy | 9 (6.1) | 15 (12.9) |  | 19 (5.8) | 18 (6.4) |  |
| Biological agents 1°line |  |  |  |  |  |  |
| Anti-EGFR use | 43 (29.4) | 30 (25.8) | 0.3873 | 86 (26.4) | 52 (18.6) | 0.378 |
| Anti-VEGF use | 72 (49.3) | 65 (56.0) |  | 176 (54.1) | 87 (31.1) |  |
| Biological agents 2°line |  |  |  |  |  |  |
| Anti-EGFR use | 9 (6.1) | 7 (6.0) | 1 | 17 (5.2) | 11 (3.9) | 0.5379 |
| Anti-VEGF use | 97 (66.4) | 81 (69.8) |  | 198 (60.9) | 100 (35.8) |  |
| Biological agents 3°line |  |  |  |  |  |  |
| Anti-EGFR use | 10 (6.8) | 10 (8.6) | 0.7089 | 13 (4.0) | 12 (4.3) | 0.747 |
| Anti-VEGF use | 4 (2.7) | 6 (5.1) |  | 9 (2.7) | 6 (2.1) |  |

Abbreviations: T, trifluridine/tipiracil; R, regorafenib; CT, chemotherapy; MSI, micro-satellites' instability; PS, performance status.

The median age was 68 years (range 30–84) in the sequential groups and 69 years (range 34–87) in the non-sequential group. The male gender and ECOG PS = 0–1 prevail with a total of 504 patients (58.1%) and 710 patients (81.9%), respectively. Patients who were ≥70 years old were prevalent only in the T group (54.7%). All of the patients had previously received a fluoropyrimidine-containing regimen, and bevacizumab was the biological agent most frequently utilized both in the first and second line, while anti-EGFR antibodies were mostly used as the third-line treatment. The liver was the only metastatic site in 106 individuals (12.2%), and the overall percentage of left-side malignancies was 68%.

The overall median follow-up period was 7.2 months (95% CI = 6.5–81.1) for all patients, being 10.8 months (95% confidence interval (CI) = 9.4–51.3) in the T/R group, 13.7 months (95% CI = 12.5–73.0) in the R/T group, 5 months (95% CI = 4.4–46.5) in the T group, and 4.2 months (95% CI = 4.2–81.1) in the R group. Patients who discontinued treatment in the sequence or not did so because of progression of metastatic colorectal cancer.

In the T/R sequence, the median treatment time was 4.1 months for trifluridine/tipiracil and 3.4 months for regorafenib. In the reverse sequence, regorafenib had a median duration of 4.3 months compared to trifluridine/tipiracil's 3.7 months.

### 3.2. Survival and Responses in Sequential Treatment Groups

In this matched sample, the R/T group showed a significantly longer median OS (mOS) (15.9 months; 95% CI = 13.9–73.0; hazard ratio (HR) = 1.41) versus the T/R group (13.9 months; 95% CI = 11.0–15.1; HR = 0.70) (*p* = 0.0194) (Figure 2A). The efficacy results are shown in Table 2. The median PFS (mPFS) was also statistically significant in favor of the R/T sequence (11.2 vs. 8.8 months, *p* = 0.0005 with R/T and T/R, respectively (Figure 2B).

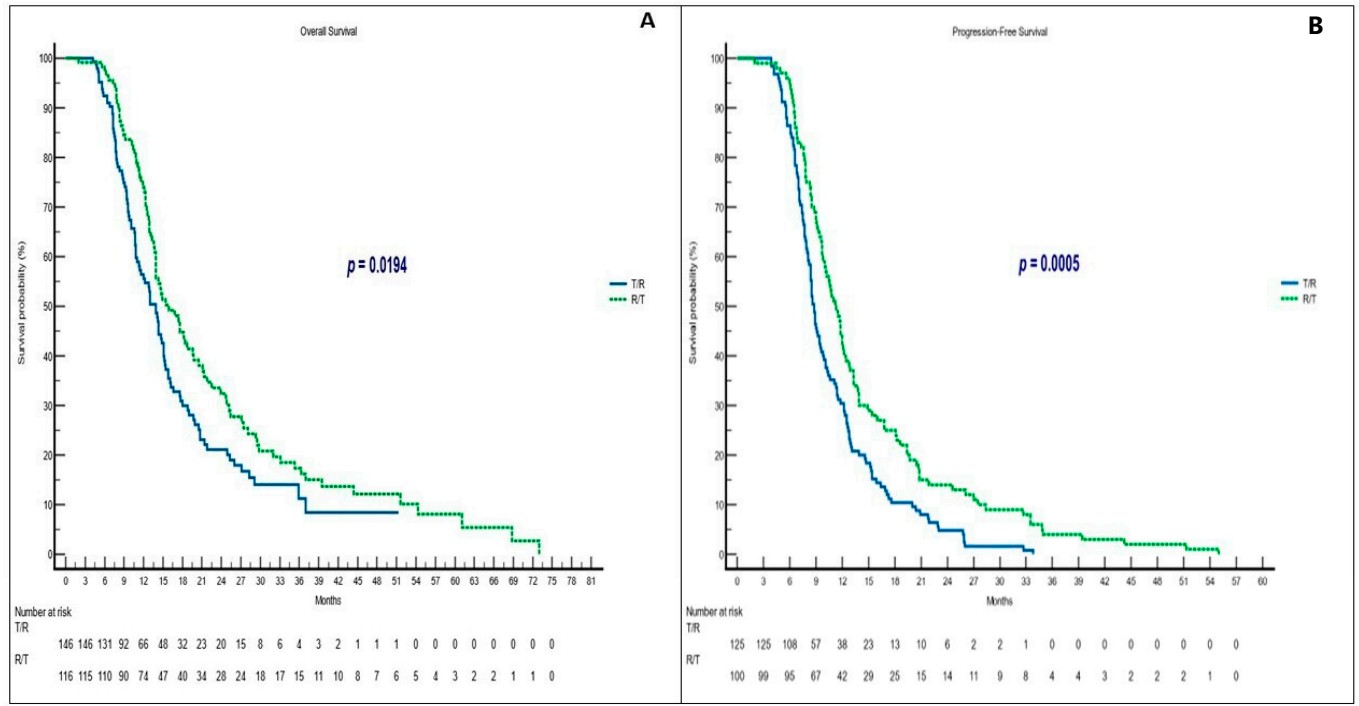

**Figure 2.** Clinical outcomes in sequential treatment. (**A**) OS in T/R and R/T groups; (**B**) PFS in T/R and R/T groups. Abbreviations: OS, overall survival; PFS, progression-free survival; T, trifluridine/tipiracil; R, regorafenib.

**Table 2.** Efficacy outcomes.

| | OS | | | | PFS | | | | ORR | | DCR | |
|---|---|---|---|---|---|---|---|---|---|---|---|---|
| | mOS (mos) | 95% CI | HR | *p*-Value | mPFS (mos) | 95% CI | HR | *p*-Value | PR + CR (%) | *p*-Value | PR + CR + SD (%) | *p*-Value |
| **T/R** | 13.9 | 11.0–15.1 | 0.70 0.52–0.94 | 0.0194 | 8.8 | 8.2–33.8 | 0.61 0.46–0.80 | 0.0005 | 5.8 | 0.5177 | 34.1 | 0.5006 |
| **R/T** | 15.9 | 13.9–73.0 | 1.41 1.05–1.89 | | 11.2 | 9.7–55.0 | 1.62 1.23–2.12 | | 3.1 | | 47.9 | |
| **T** | 6.3 | 5.6–7.4 | 1.10 0.92–1.32 | 0.2860 | 3.1 | 3.0–44.7 | 0.87 0.73–1.04 | 0.1435 | 2.5 | 1 | 23.3 | 0.6793 |
| **R** | 5.1 | 4.5–81.1 | 0.90 0.75–1.08 | | 3.0 | 2.9–54.1 | 1.14 0.95–1.36 | | 2.3 | | 21.7 | |

Abbreviations: OS, overall survival; mOS, median overall survival; PFS, progression-free survival; mPFS, median progression-free survival; 95%CI, 95% confidence interval; HR, hazard ratio; T, trifluridine/tipiracil; R, regorafenib; PR, partial response; CR, complete response; SD, stable disease; ORR, overall response rate; DCR, disease control rate.

In an exploratory subgroups analysis, we identify a correlation between the R/T sequence and longer mOS and mPFS for patients younger than 70 years (*p* = 0.1201 and

*p* = 0.0084, respectively), males (*p* = 0.0547 and *p* = 0.0027, respectively), wt RAS tumors (*p* = 0.0635 and *p* = 0.0022, respectively), ECOG PS = 0–1 (*p* = 0.0296 and *p* = 0.0038, respectively), and in patients with liver metastases only (*p* = 0.0175 and *p* = 0.0191, respectively) (Figures 3 and 4).

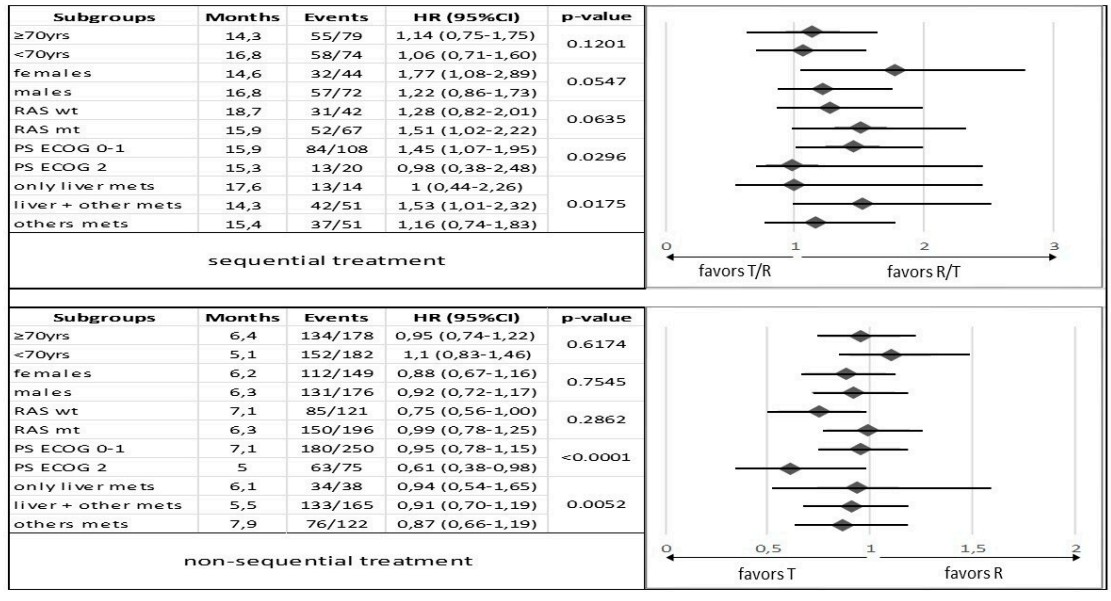

**Figure 3.** Subgroup analysis comparing overall survival between patients in the sequential treatment and in the non-sequential treatment with trifluridine/tipiracil and regorafenib groups stratified by baseline characteristics. Abbreviations: HR, hazard ratio; PS, performance status; T, trifluridine/tipiracil, R, regorafenib; mets, metastases; wt, wild type; mt, mutant type; 95% CI, 95% confidence interval; yrs, years.

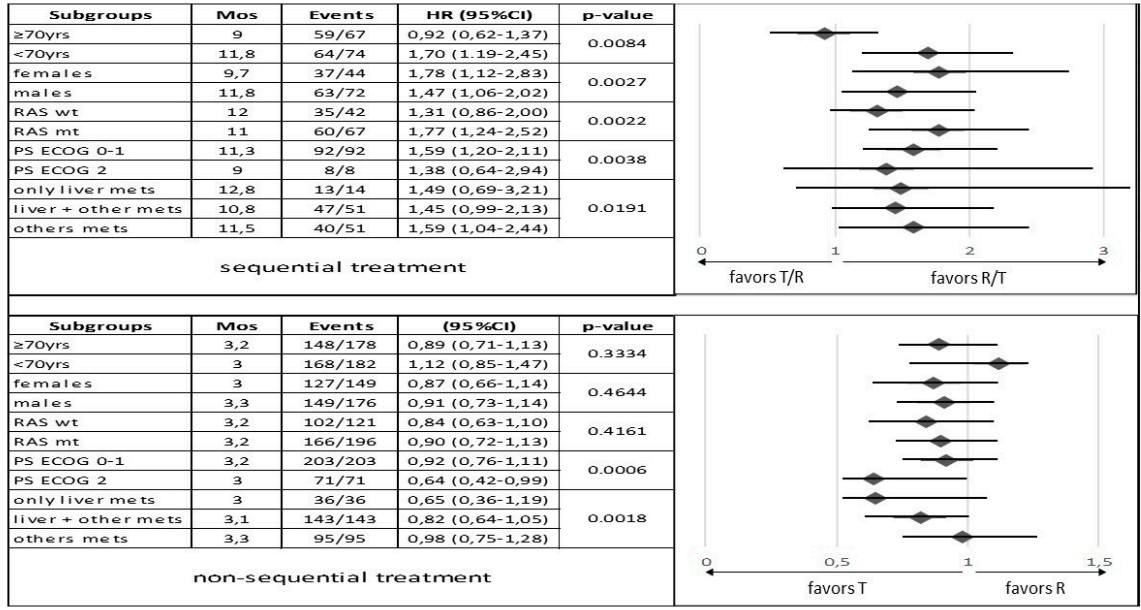

**Figure 4.** Subgroup analysis comparing progression-free survival between patients in the sequential treatment and in the non-sequential treatment with trifluridine/tipiracil and regorafenib groups stratified by baseline characteristics. Abbreviations: HR, hazard ratio; PS, performance status; T, trifluridine/tipiracil, R, regorafenib; mets, metastases; mos, months; wt, wild type; mt, mutant type; 95% CI, 95% confidence interval; yrs, years.

No complete responses were observed. The R/T group, however, showed a higher disease control rate (47.9% vs. 34.1%) than the T/R group (*p* = 0.0506), particularly in patients with wt RAS tumors (51.5%; *p* = 0.0009) and liver metastases only (72.7%; *p* = 0.0693).

The partial responses rate was 5.8% in the T/R sequence vs. 3.1% in the R/T group (*p* = 0.5177).

### 3.3. Survival and Responses in Non-Sequential Treatment Groups

No significant differences in clinical outcomes (mOS and mPFS) were observed between patients receiving T or R alone, which supports the evidence from the literature [6,8,17,26,28,35,42] (Figure 5A,B). A statistically significant difference in mOS was observed in subgroup analysis for patients in the T group with non-hepatic metastases (*p* = 0.0052) and ECOG PS = 0–1 (*p* = 0.0001). No significant change in mPFS according to subgroups was seen, with the exception of patients with ECOG PS = 0–1 in the T group with *p* = 0.0006 (Figures 3 and 4).

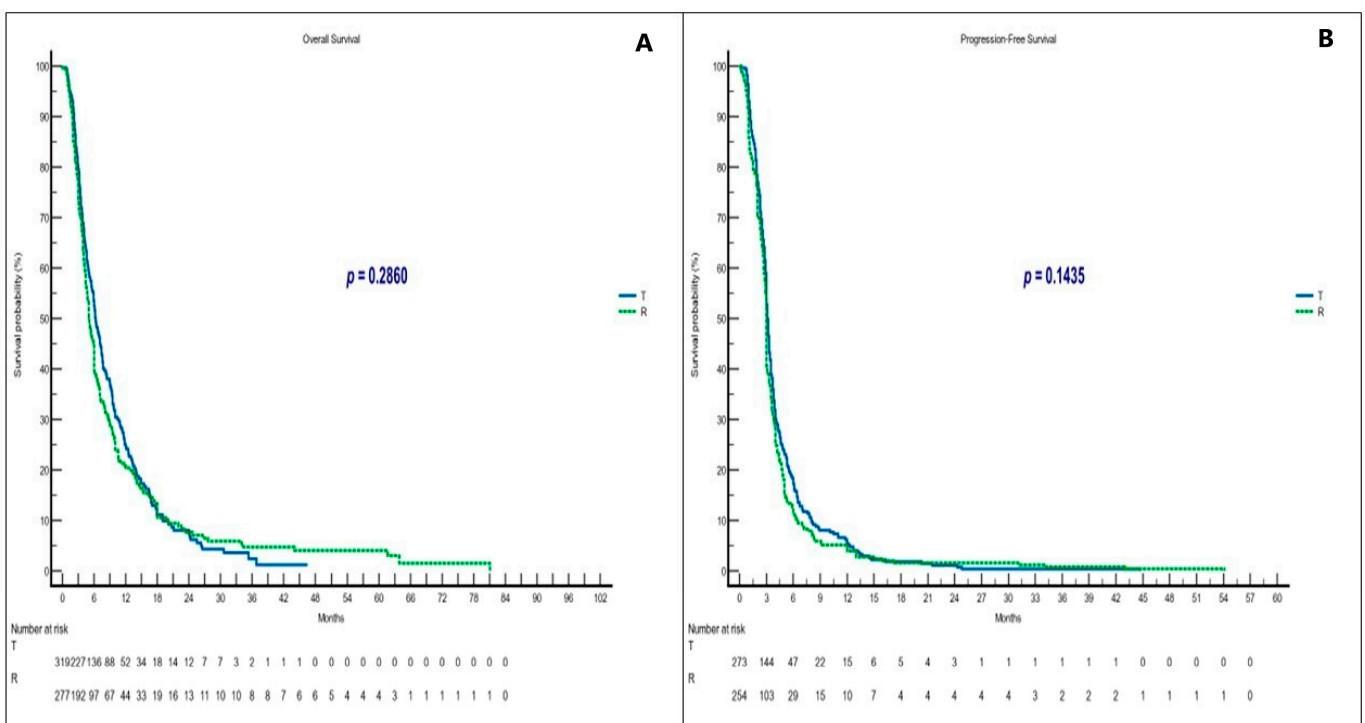

**Figure 5.** Clinical outcomes in non-sequential treatment. (**A**) OS in T and R groups; (**B**) PFS in T and R groups. Abbreviations: OS, overall survival; PFS, progression-free survival; T, trifluridine/tipiracil; R, regorafenib.

The ORR and DCR for T and R were also comparable (Table 2). Sex (men) (*p* = 0.0080) and ECOG PS = 0–1 (*p* = 0.0169) showed a significant improvement in DCR (25% and 26.2%, respectively), regardless of treatment.

Regarding the effectiveness of the treatment, we found a trend toward better ORR (3.4%) only in patients who had an ECOG PS = 0–1 (*p* = 0.0456) in the T group and a trend toward better DCR in the R group patients with liver metastases only (34.3%) (*p* = 0.0289). Moreover, we found a small but statistically significant advantage in DCR (23.6%) in favor of patients less than 70 years old receiving T vs. the R group patients (23.6 vs. 22.4%, *p* = 0.0487). Furthermore, patients in the T group with an mt RAS tumor and non-hepatic metastases had the highest DCRs (34.3% and 27.4%, respectively). Only one complete response was reported in the R group, in a 70-year-old female patient with wt RAS, ECOG PS = 0–1, and non-hepatic metastases.

*3.4. Safety*

Table 3 summarizes drug-related toxicities. Several dosing schedules of R are employed in the clinic to ameliorate toxicities.

**Table 3.** Drug-related toxicities.

| | T/R | | R/T | | | T | | R | | |
| --- | --- | --- | --- | --- | --- | --- | --- | --- | --- | --- |
| | N | % | N | % | *p*-Value | N | % | N | % | *p*-Value |
| All events G3/G4 N = 582 | 155 | 26.6 | 132 | 22.6 | - | 160 | 27.4 | 135 | 23.1 | - |
| All pts who experienced G3/G4 toxicities N = 390 | 89/146 | 60.9 (22.8) | 79/116 | 68.1 (20.2) | 0.24 | 118/325 | 36.3 (30.2) | 104/279 | 37.2 (26.6) | 0.86 |
| All Haematologic events G3/G4 N = 289 | 87 | 56.1 (14.9) | 65 | 49.2 (11.1) | 0.28 | 126 | 78.7 (21.6) | 11 | 8.1 (1.8) | <0.00001 |
| All non Haematologic events G3/G4 N=293 | 68 | 43.8 (11.6) | 67 | 50.7 (11.5) | | 34 | 21.2 (5.8) | 124 | 91.8 (21.3) | |
| **Most common Haematologic toxicities G3/G4** | | | | | | | | | | |
| Neutropenia | 68 | 78.2 | 43 | 66.2 | 0.13 | 82 | 65.1 | 3 | 27.3 | 0.02 |
| Leucopenia | 2 | 2.3 | 0 | 0 | 0.5 | 6 | 4.8 | 0 | 0 | 1 |
| Thrombocytopenia | 3 | 3.4 | 5 | 7.7 | 0.28 | 3 | 2.4 | 0 | 0 | 1 |
| Anemia | 14 | 16.1 | 17 | 26.2 | 0.15 | 35 | 27.8 | 8 | 72.7 | 0.0042 |
| **Most common non Haematologic toxicities G3/G4** | | | | | | | | | | |
| Fatigue | 33 | 48.5 | 19 | 28.4 | 0.02 | 21 | 61.8 | 34 | 27.4 | 0.0004 |
| Hand-foot skin reaction | 5 | 7.4 | 25 | 37.3 | 0.01 | 0 | 0 | 42 | 33.9 | 0.01 |
| Liver dysfunctions | 2 | 2.9 | 2 | 3 | 1 | 2 | 5.9 | 1 | 0.8 | 0.11 |
| Diarrhea | 10 | 14.7 | 2 | 3 | 0.03 | 7 | 20.6 | 13 | 10.5 | 1.14 |
| Skin disorders | 2 | 2.9 | 3 | 4.5 | 1 | 0 | 0 | 11 | 8.9 | 0.12 |
| Others | 16 | 23.5 | 16 | 23.9 | 1 | 4 | 11.8 | 23 | 18.5 | 0.44 |

Abbreviations: T, trifluridine/tipiracil; R, regorafenib.

Regardless of treatment group, 541 patients were given R; of these, 263 (48.7%) began treatment at an initial dose of 80 mg per day (ReDos schedule), 58 (10.7%) at 120 mg per day, and 220 (40.6%) at 160 mg per day ($p = 0.0747$).

Overall, 390 patients experienced 582 grade 3/4 toxicities; in detail, 155 events (26.6%) were recorded in 89 patients in the T/R group; 132 (22.6%) were recorded in 79 patients in the R/T sequence group; 160 (27.4%) were recorded in 118 patients in the T group; and 135 (23.1%) were recorded in 104 patients in the R group.

The frequency of grade 3/4 hand-foot skin reactions (HFSR) was lower in the T/R sequence compared to the R/T sequence ($p = 0.01$), while the incidence of grade 3/4 neutropenia in the R/T group was slightly lower than in the T/R group ($p = 0.13$).

Grade 3/4 hematologic toxicity was most common in patients who underwent the non-sequential treatment (126 events in the T group), while non-hematologic grade 3/4 AEs were prevalent in the R group (124 events). There was no therapy-related death recorded.

Trifluridine/tipiracil dose reduction due to toxicity was prevalent in the non-sequential group as follows: 47.3% of patients at 30 mg/m$^2$; 44.8% at 25 mg/m$^2$ and 55% at 20 mg/m$^2$.

Regorafenib was also reduced in the non-sequential arm as follows: 62.6% of patients at 120 mg/day and 54.9% at 80 mg/day. The dosage reductions of T and R in the sequential groups were lower.

## 4. Discussion

Our multicentric and retrospective study investigated the safety and efficacy of R and T in routine clinical practice as a third-line treatment, or greater, for mCRC.

We also searched for the benefits and drawbacks of sequential and non-sequential therapy with these two drugs. To our knowledge, there are no other large retrospective studies that have examined the efficiency and safety of sequential administration of these two drugs. Similar published studies all have smaller sample sizes, although they have shown long OS and PFS [26,27].

The findings of this study are important because it is unlikely that these drugs will ever be directly tested in a prospective head-to-head trial. At least five meta-analyses

of prospective studies comparing the results of patients using R or T have been carried out. No statistically significant differences in OS or PFS were found in any of these meta-analyses [6,24,43–45]. A number of modest retrospective investigations employing real-world populations in Asia [8,25–27,34] and Europe [46] have similarly failed to show any discernible difference in efficacy between these two treatments.

In an initial report of this study, on a smaller sample size (49 patients) we observed that patients receiving sequential treatment were still suitable for additional systemic therapies and can achieve a high disease control rate [36].

The results achieved in this larger retrospective study could be a starting point for additional discussion. In fact, we believe that at the present time, one of the most interesting issues in the third-line treatment of colorectal cancer is which agents should be administered first. We thought that our multicenter experience, despite the limits of the retrospective analysis, provides a further step in clarifying this issue. The most relevant result is a statistically significant difference in mOS (15.9 vs. 13.9 months, $p = 0.0194$, for R/T vs. T/R, respectively) and in mPFS (11.2 vs. 8.8 months, $p = 0.0005$, for R/T vs. T/R). Disease control is also a crucial element when the treatment of refractory mCRC reaches the third line, and we found that the R/T sequence was associated with significantly improved DCR (47.9% R/T vs. 34.1% T/R; $p = 0.0506$).

There is a strong need to uncover relevant clinical traits or biomarkers that could predict responsiveness to these two agents. Prior research has revealed that clinical factors such as age and prior targeted treatment use could be predictors of response to T or R [16,34,43–45]. We also performed various subgroup analyses in the wake of prior retrospective studies in order to identify potential prognostic markers. So far (and the data are entirely preliminary and debatable), we also detected a statistical trend toward a better prognosis with R/T sequential therapy vs. the T/R sequence, in patients with wt RAS tumors, ECOG PS = 0–1, hepatic metastases only, in men and in patients younger than 70 years. In other words, our analysis suggests that the R-to-T sequence could be recommended for patients diagnosed with refractory mCRC in the setting of better performance status, wt RAS, only liver metastases, in men and finally in patients <70 years old. It would be useful to confirm this by carrying out prospective studies.

In terms of safety, we have noticed that when using a sequential therapeutic approach, even when the drug was administered after T, the incidence of HFSR in patients treated with R was not noticeably elevated, and vice versa, patients treated with T who had previously received R showed a reduced incidence of neutropenia. This finding should be regarded with caution due to the retrospective nature of our study. In this regard, for example, it should be noted that in clinical practice, it is difficult to distinguish between fatigue caused by a drug and the progression of the disease itself.

As already known from the literature, no differences in OS and PFS between T and R given in a non-sequential manner have been observed [6,8,10,12,16,17,25–27,44,45]. The most recent evidence in favor of the use of T and R comes from randomized controlled trials with best supportive care and a placebo as the control. The OS improvement shown in these trials was not substantial. In the CORRECT trial, regorafenib, when compared with placebo and best supportive care, increased the median OS from 5.0 to 6.4 months ($p = 0.005$) [9]. Similarly to this, in the RECOURSE study, trifluridine/tipiracil increased the median OS from 5.3 to 7.1 months ($p \leq 0.001$) when compared with best supportive care alone [13]. The small percentage of patients in both of these trials achieving an objective response rate (<5%) may be to blame for the minimal OS benefit. This finding indicates that the main therapeutic effect of T and R is disease control. In our analysis, patients treated with either R or T achieved a median OS of 6.3 and 5.1 months, respectively, and this is comparable to those shown in clinical studies, leading to the FDA approval of the drugs. This shows that a real-world population does not disregard the small benefits of these agents. The results of our study are consistent with previous meta-analyses and either with studies that compared T and R.

To date, only one retrospective investigation conducted in Japan has evidenced a difference in survival outcomes in patients receiving R and T non-sequentially, with a statistically significant rise in OS among patients receiving T [29].

We also observed a statistically significant prognosis for patients with non-hepatic metastases ($p = 0.0052$) and ECOG PS = 0–1 ($p \leq 0.0001$) in the T group.

The findings of our study do not align with previously published reports regarding the necessity of treatment reductions or interruptions [9,32]. Although a ReDos protocol was prescribed for 48.7% of patients in our trial, contrary to our expectations, the dose reductions of T were smaller than those of R. Both R and T can be regarded as a financially viable therapy for refractory mCRC, while trifluridine/tipiracil appears to be less expensive. It is possible to turn the situation around and make R more cost-effective than T by using a dose-escalation method [47]. However, in the sequential groups, the lowered doses were smaller. Although initiation of treatment with a reduced dose is not officially recommended, regorafenib dose-optimization can be considered a reasonable choice according to the ReDOS study [9,32].

Regarding drug safety, the toxicity profile that we observed replicates the results of previous retrospective clinical investigations [9,12,16]. In the non-sequential setting, the prevalence of hematological AEs during T treatment compared to the non-hematological AEs most represented in R-treated groups was confirmed. Bone marrow suppression, particularly neutropenia, and anemia (65.1% and 27.8%, respectively) were major adverse events seen in the T group. Hand-foot syndrome and weariness (33.9% and 27.4%, respectively) were the two most notable serious adverse events in the R group.

Although multicentric and extensive, the principal drawback of this study is surely its retrospective nature. We have also undoubtedly neglected to evaluate additional subgroups that might be the subject of further research. We made an effort to determine the best methods for administering T and R in treatment for mCRC beyond the second line in terms of efficacy and safety, but more studies are needed. Further trials are underway to confirm the benefit of T combined with bevacizumab and to investigate the combination of T or R with other targeted treatments or immunotherapy. We are confident that these investigations will lead to additional improvements in the outcomes of patients with mCRC.

## 5. Conclusions

In conclusion, our analysis proves the effectiveness and manageable adverse effects of late-line administration of T and R for the treatment of mCRC in a real-world setting.

According to our research, administering R prior to T can help to prolong both PFS and OS, even without tumor shrinkage. Although it is not yet established which agent should be administered first, we suggest that some predictors such as ECOG PS, RAS status, and site of metastases (liver only) could be determining factors for survival in the R/T sequence. Future research is needed to define the optimal order of administration of these two drugs, validate the effective starting dose of R, and explore the efficacy of sequential (T/R or R/T) treatment combined with molecular-targeted drugs. We are optimistic that these investigations will result in even better outcomes for people with mCRC.

**Author Contributions:** Conceptualization, C.S.; methodology, C.S.; software, C.S.; validation, C.S.; formal analysis, C.S.; investigation: C.S., M.A.C., M.B., A.A., J.L., A.M., L.A., I.V.Z., M.S., M.G.C., C.M., E.D., A.C., D.G., M.R., A.E., D.C.C., G.A., F.M., F.Z., M.G.M., F.S. and R.S.; resources, C.S.; data curation, C.S.; Writing—Original draft preparation, C.S.; Writing—Review and editing, C.S. and E.M.R.; visualization, C.S.; supervision, C.S. and E.M.R.; project administration, C.S. All authors have read and agreed to the published version of the manuscript.

**Funding:** This research received no external funding.

**Institutional Review Board Statement:** The study was conducted in accordance with the Declaration of Helsinki, and approved by the Ethics Committee of Lazio 1 (protocol code 1021, 09/29/2022).

**Informed Consent Statement:** Patient consent was waived due to the nature of the retrospective investigation.

**Data Availability Statement:** The data to support the results reported in this study are available from the corresponding author on reasonable request.

**Conflicts of Interest:** The authors declare no conflict of interest.

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
