# Peer review of "Treatment Settings and Outcomes with Regorafenib and Trifluridine/Tipiracil at Third-Line Treatment and beyond in Metastatic Colorectal Cancer: A Real-World Multicenter Retrospective Study"

_curroncol, doi:10.3390/curroncol30060413_

Round 1

Reviewer 1 Report

The authors presented about treatment settings and outcomes with regorafenib and TAS-102 at 3rd-line and beyond in mCRC by a retrospective study. Among the patients who received both of regorafenib and TAS-102, the efficacy outcomes in the regorafenib first group were more improvement than those in the TAS-102 first group. Among the patients who received regorafenib or TAS-102 only, there were not significant differences in outcomes between the two groups.

This study is interesting for me, but I think that these results misunderstand, especially in the results of R/T vs T/R.

1) I think that the authors should try to eliminate bias as much as possible when the retrospective data use. But, this study did not been performed.

2) The comparison between R/T and T/R was performed in patients who had be able to receive these two drugs. The authors showed some predictors like ECOG PS, RAS status, and site of metastasis could be determining factors for survival in the R/T sequence. In the case of clinical setting, however, it is unknown whether the patients with those predictors will be able to receive two drugs before receiving R or T treatment. Thus, I think those predictors don't have meaningful as predictors.

3) Why did the patients discontinued initial or sequence treatment?  Progressive disease? I think that these pieces of information are important to the readers. 

4) Although PFS was defined as the time between the first treatment and event during the second treatment, the authors should consider making corrections such as changing to PFS2. 

5) Are the P-values shown in Figure 3 for interactions? The authors should show about this in the Statistical analysis section.

6) Are there enough patient background factors if the comparison between two drugs perform? The authors should consider various predictive factors which has been reported in the previous studies.

Author Response

Dear Reviewer 1

Thank you so much for your comments

I really appreciate

Carlo Signorelli

Reviewer 2 Report

The present study by Signorelli C et al is very interesting and addressing a crucial question regarding 3rd line and above treatments for mCRC. It is also important that it is a real-world study providing evidence more representative to general population compared to clinical studies. 

My main methodological concern in the study is the definition of PFS in sequential treatment. As per authors "it is defined as the time between the first treatment, T or R, and disease progression or death during the second treatment". I am not aware of any other studies using this approach. This definition is not very specific regarding the actual effect of each treatment and the results may be misinterpreted. Furthermore, the authors are not even providing descriptive information regarding the duration of each treatment. This will also be useful to assess the effect of post-progression OS (e.g. results in favor of R/T may be just due to the effect of longer post-progression OS in patients receiving T after R; in the single treatment groups patients on T presented aprox. 24% longer OS compared to R group)

1. Is there any reason that authors used this specific definition of PFS in sequential treatment? Any references?

2. Please provide the duration of each treatment in the sequential treatment groups.

3. Please sevise abstract to be more clear that comparing between 2 sequential treatment groups and between 2 single treatment groups and not sequential with single treatment groups. 

Minor edits needed

Author Response

Dear Reviewer 2

Thank a lot for your comments

Carlo Signorelli

Reviewer 3 Report

This is a very well written paper.

Despite the retrospective nature, the topic and findings are important for patients with metastatic colorectal cancer who have gone beyond standard chemotherapy.

There is not much data in this area and it reinforces the importance of this study.

Author Response

Dear Reviewer 3,

Thank you so much for your comments.

I really appreciate.

Carlo Signorelli

Round 2

Reviewer 1 Report

I am sorry, but I cannot accept agree with your answers.

Point 1: I understand your research has some limitations including bias. Thus, you should try to eliminate bias as much as possible when the retrospective data use.

Point 2: You wrote in the discussion section, "we also detected a statistical trend toward a better prognosis with R/T sequential therapy vs. the T/R sequence, in patients with wt RAS tumors, ECOG PS=0-1, hepatic metastases only, in men and in patients younger than 70 years. Said another way, our analysis suggests that the R-to-T sequence could be recommended for patients diagnosed with refractory mCRC in the setting of better performance status, wt RAS, only liver metastases, in men and finally in patients <70 years old. It would be useful to confirm this by carrying out prospective studies." However, from Figure 3 and 4, I understood that these predictors based on results comparing R/T sequence with T/R sequence. If so, firstly, you must find significant favorable factors to patients who received two drugs compared to patients who received only R or T treatment by multivariate analysis. Because it is unknown whether the patients with those predictors will be able to receive two drugs before receiving R or T treatment. Next, among patients with favorable factors, you should compare R/T with T/R by multivariate analysis.

Point 3: You replayed, "Yes, you are right. Patients who discontinued treatment in the sequence or not, did so because of progression of metastatic colorectal cancer. I take this opportunity to add this information to the manuscript. Thank you very much.", but I could not find where the sentences added.

Point 4: You revised only the definition of PFS2. As the definitions are revised, the results should also be revised. You wrote in the Results section, "The median PFS (mPFS) was also statistically significant in favor of the R/T sequence (11.2 vs. 8.8 months, p=0.0005 with R/T and T/R respectively. (Figure 2B).", but these PFSs were too long. Are these PFS2? If so, Table 2 is also wrong.

Point 5: You could not answer for my comment. For example, the top two rows in the Figure 3, HR=1.14 showed T/R versus R/T among ≥70 years patients, and HR=1.06 showed  T/R versus R/T among <70 years patients. Was not p-value=0.1201 result of interaction test for these two relationships?

Author Response

Point 1: I understand your research has some limitations including bias. Thus, you should try to eliminate bias as much as possible when the retrospective data use.

Response 1: You are entirely correct, Reviewer. You can believe that I made every effort to minimize bias when I processed the data and wrote the manuscript. I believe that removing and preventing any form of bias in a retrospective study like this is a very challenging issue. All patients receiving R or T or sequential R/T and T/R were included in the current study to remove potential selection bias. A predetermined investigator who was blind to the study's results and outcomes was responsible for selecting the patients. Patients were not chosen by the principal investigator who was in charge of statistical analyses. To reduce the possibility of distortion bias, endpoints were predetermined at the time of the studies. We should be aware that the provided results should be viewed as exploratory because of retrospective nature of the study.

Point 2: You wrote in the discussion section, "we also detected a statistical trend toward a better prognosis with R/T sequential therapy vs. the T/R sequence, in patients with wt RAS tumors, ECOG PS=0-1, hepatic metastases only, in men and in patients younger than 70 years. Said another way, our analysis suggests that the R-to-T sequence could be recommended for patients diagnosed with refractory mCRC in the setting of better performance status, wt RAS, only liver metastases, in men and finally in patients <70 years old. It would be useful to confirm this by carrying out prospective studies." However, from Figure 3 and 4, I understood that these predictors based on results comparing R/T sequence with T/R sequence. If so, firstly, you must find significant favorable factors to patients who received two drugs compared to patients who received only R or T treatment by multivariate analysis. Because it is unknown whether the patients with those predictors will be able to receive two drugs before receiving R or T treatment. Next, among patients with favorable factors, you should compare R/T with T/R by multivariate analysis.

Response 2: In an attempt to identify potential clinical features or biomarkers that may be predictive of response to trifluridine/tipiracil and regorafenib administered sequentially or alone, I performed a subgroup analysis, illustrated with forest plots in Figures 3 and 4. We have analysed the same clinical characteristics in both sequential and non sequential treatment. In this analysis our study attempted to give an indication or identify an option before choosing to administer trifluridine/tipiracil or regorafenib sequentially or not. Obviously it is clinically important to identify the patient who is fit enough to receive both drugs or just one of the two drugs. If the patient is fit for sequential treatment, the variables listed in figures 3 and 4 have shown a correlation, significant or not, with an outcome such as OS and PFS and attempt to give us an indication of which drug to administer first in the sequence. The same figures show the characteristics of the patient that could best relate to a better OS and PFS in the patient to whom will be administered or just trifluridine/tipiracil or just regorafenib. However, we are unable to determine if these traits are prognostic or predictive, despite the fact that we have shown that patients with these characteristics have better results. This is what I was able to observe in this retrospective study born from daily clinical practice. Comparing T/R and R/T sequences by multivariate analysis among patients with favourable factors could be further explored. Due to the lack of any clear clinical or biomarker characteristics to predict response to treatment with trifluridine/tipiracil or regorafenib, the choice of agent for suitable patients should be individualised, with particular attention to differences in toxicity and schedule of administration of each drug.

Point 3: You replayed, "Yes, you are right. Patients who discontinued treatment in the sequence or not, did so because of progression of metastatic colorectal cancer. I take this opportunity to add this information to the manuscript. Thank you very much.", but I could not find where the sentences added.

Response 3: Sorry, you’re right. I just added the information you requested. 

Point 4: You revised only the definition of PFS2. As the definitions are revised, the results should also be revised. You wrote in the Results section, "The median PFS (mPFS) was also statistically significant in favor of the R/T sequence (11.2 vs. 8.8 months, p=0.0005 with R/T and T/R respectively. (Figure 2B).", but these PFSs were too long. Are these PFS2? If so, Table 2 is also wrong.

Response 4: I've updated the PFS2 definition. For patients who had undergone the crossover between therapies following a first progression, PFS2 is defined as the interval from the start of secondary treatment to the second progression. The time between the first treatment, T or R, and the progression of the disease or death during the second treatment, R or T, is how I would define PFS in sequential treatment in the original draft of my manuscript's "Patients and Methods" section, as opposed to PFS2, which is a different concept. I made an effort to use the straightforward definition of PFS while indicating the context in which it is employed. So when I refer to PFS in the context of sequential therapy, I mean the definition given above, and when I refer to PFS in the context of non-sequential treatment, I mean the period of time from the beginning of treatment until the onset of the disease or death. I believe that utilizing the word PFS2 within the paper may lead to confusion. I implore you to take into consideration the definition of PFS as it was in the original version of the manuscript because it is defined every time we discuss PFS in the text, tables, and figures, whether it be PFS of sequential treatment or PFS of non-sequential treatment. The median PFS, not the PFS2, displayed in Figure 2B, is longer and in favor of the R/T sequence since it represents the period between two treatments. Please excuse me, but after careful consideration, rereading articles from the literature, and fearing that I might upset the study's design, I decided to remove the term PFS2 because I feel that it could cause confusion and prevent people from understanding the concept I'm trying to convey. I used the same simple definition of PFS in sequential treatment in a small experience published in 2021 (Signorelli C, et al. Regorafenib-to-trifluridine/tipiracil Versus the Reverse Sequence for Refractory Metastatic Colorectal Cancer Pts: A Mul-ticenter Retrospective Real-Life Experience. Anticancer Res. 2021 May;41(5):2553-2561. doi: 10.21873/anticanres.15033). I sincerely hope you comprehend what I'm trying to say.

Point 5: You could not answer for my comment. For example, the top two rows in the Figure 3, HR=1.14 showed T/R versus R/T among ≥70 years patients, and HR=1.06 showed  T/R versus R/T among <70 years patients. Was not p-value=0.1201 result of interaction test for these two

Response 5: The P-values displayed in Figure 3 are not the result of the interaction test for the coupled variables. These P-values were calculated using the Kaplan-Meier method to estimate the OS for each subgroup.